# Spin-Trapping Analysis of the Thermal Degradation Reaction of Polyamide 66

**DOI:** 10.3390/polym14214748

**Published:** 2022-11-05

**Authors:** Akihiro Kurima, Kenji Kinashi, Wataru Sakai, Naoto Tsutsumi

**Affiliations:** 1Doctor’s Program of Materials Chemistry, Graduate School of Science and Technology, Kyoto Institute of Technology, Matsugasaki Sakyo, Kyoto 606-8585, Japan; 2Platform Laboratory for Science & Technology, Asahi-Kasei Corporation, Moriyama City 524-0002, Japan; 3Faculty of Materials Science and Engineering, Kyoto Institute of Technology, Kyoto 606-8585, Japan

**Keywords:** polyamide 6,6 (PA66), nylon 66, thermal degradation, electron spin resonance (ESR), spin-trapping method, radical intermediates, 1,3,5-tri-*tert*-butyl-2-nitrosobenzene (TTBNB)

## Abstract

The radical mechanisms of the thermal degradation of polyamide 66 (PA66) occurring under a vacuum at a temperature range between 80 °C and 240 °C (which includes the temperature of practical applications) were investigated using a spin-trapping electron spin resonance (ST-ESR) technique, as well as FTIR, TG-DTA, and GPC methods. No significant weight loss and no sign of thermal degradation are observed at this temperature range under oxygen-free conditions, but a slight production of secondary amine groups is confirmed by FTIR. GPC analysis shows a small degradation by the main chain scission. ST-ESR analysis reveals two intermediate radicals which are produced in the thermal degradation of PA66: (a) a ^●^CH_2_− radical generated by main chain scission and (b) a −^●^CH− radical generated by hydrogen abstraction from the methylene group of the main chain. The ST-ESR result does not directly confirm that a −NH−**^●^**CH− radical is produced, although this reaction has been previously inferred as the initiation reaction of the thermal degradation of PA; however, the presence of −^●^CH− radicals strongly suggests the occurrence of this initiation reaction, which takes place on the α-carbon next to the NH group. The ST-ESR analysis reveals very small levels of reaction, which cannot be observed by common analytical methods such as FTIR and NMR.

## 1. Introduction

Degradation is an unavoidable problem affecting resin materials used in industry. For example, microplastics are a worldwide environmental problem [1,2,3] caused by the fragmentation of plastics as a result of degradation [4,5]. Plastic materials are cheaper, lighter, and easier to form than metals and ceramics, which have been used since ancient times. However, plastics are far less resistant to light and heat than these materials, thus, anti-degradation agents are commonly added to plastics to extend their life. It is desirable to select the best anti-degradation agent based on the degradation mechanism for each material; however, in practice, in material development, different anti-degradation agents are often tested through trial and error based on experience. This approach does not always result in a logical solution based on the degradation mechanism that actually affects the materials. One of the reasons for this shortcoming is that degradation is made up of multi-step reactions, including chain reactions, which makes the analysis of the reaction mechanism complex and difficult. Moreover, as many textbooks and papers assert, the degradation of polymer materials proceeds through radical reactions, which, however, cannot be observed by common analytical methods such as FTIR and NMR. Generally, reaction analysis is performed by measuring a sample after degradation, using, for example, Fourier transform infrared spectroscopy (FTIR). However, with this method, it is necessary to estimate the radical reaction path from the final products after degradation, and thus, the exact path, including radical species, may not be known directly. In addition, due to the sensitivity limits of the analysis, changes in chemical structures cannot be detected unless the sample has been significantly degraded, which makes the estimation of the initial reaction pathway particularly difficult.

Generally, degradation reactions are caused by light, heat, and mechanical fatigue, and most proceed through radical reactions that break polymer chains. Among the many analytical methods, electron spin resonance (ESR) is the only method that can detect radicals, and its high sensitivity enables the capture of the initial stage of degradation. However, because organic radicals are generally very unstable and their lifetimes are often counted in milliseconds or less, the detection of the radical intermediates of polymer degradation by ESR is also difficult [6,7,8]. There are methods for dealing with short-lived species using pulsed ESR measurements and cryogenic cooling methods with suppression of reaction processes, but each method has its own problems in that the equipment is extremely expensive, the species observed are limited, and the measurement conditions do not reflect the actual degradation environment.

To address these serious problems, our laboratory focused on the spin-trap method to convert short-lived organic radicals into stable radicals [9,10,11,12]. The pre-added spin-trapping reagent captures free radicals to form persistent radicals called spin adducts, which are detectable by standard ESR equipment using a continuous-wave method. The detailed analysis of the ESR spectra of the spin adducts allows the intermediate radicals to be determined as they originate from thermal degradation reactions in various polymer materials, such as poly(butylene terephthalate) [13], thermoplastic elastomer [14], poly(vinyl alcohol) [15], and poly(propylene) [16].

In this study, we attempted to analyze the radicals generated in the thermal degradation of polyamide 6,6 (PA66), a common engineering plastic that is also known as nylon 66, using a spin-trapping ESR (ST-ESR) technique. Polyamide (PA) is widely used in automobile engine parts because of its excellent heat resistance. However, there is a need to further improve its heat resistance so that it can be used in more severe conditions over a longer period. The mechanism of thermal degradation of PA66 has been studied since the 1950s. Subsequent studies, however, have yielded different results depending on degradation conditions; these studies have been reviewed in detail by Levchik, et al., in 1999 [17], and elsewhere [18,19,20]. These differences come about because there are many secondary reactions involved in the thermal degradation of PA66; as a result, the thermal degradation mechanism of PA66 is not well understood. However, most reports agree that degradation begins with hydrogen abstraction or detachment from the methylene group at the β-position of the carbonyl group (−CH_2_−CH_2_−CO−NH−) or adjacent to the NH (−CH_2_−NH−CO−). That said, these analyses are based on decomposition at high temperatures above 350 °C or conventional processing temperatures above melting point [21,22]; there are few reports on degradation at temperatures around 200 °C or lower than the melting point of PA66 [23,24,25], to which automotive and other practical parts are routinely exposed. Although the decomposition temperature at which PA shows an obvious decrease in its molecular weight is 400 °C or higher, the thermal oxidative degradation of PA66 progresses at lower temperatures, as evidenced by the decline in its physical properties when a 100 μm thick film is held at 100 °C for a long period [25]. Therefore, it is very important to understand the details of the degradation mechanism of PA66 at lower temperatures of around 200 °C or less. Although a few cases using radical generation by γ-irradiation or mechanochemistry have been analyzed [26,27,28], no studies of radicals have examined the thermal degradation of PA because intermediate radicals are quickly quenched at or above room temperature. Thus, in this paper, the thermal degradation of PA66 molecules was studied using ST-ESR by analyzing radical generation during the heating processes, firstly under oxygen-free conditions.

## 2. Materials and Methods

### 2.1. Materials

The materials used for the spin-trapping experiments presented in this study are shown in Figure 1. PA66 was synthesized by polycondensation from the nylon salts constituted with adipic acid and hexamethylenediamine at 285 °C under a vacuum of 20 Pa and then purified by reprecipitation with 1,1,1,3,3,3-hexaflouro-2-isopropanol (HFiP)/methanol. The weight average molecular weight, *M*_w_, of the obtained PA66 measured by GPC was 43,600. The spin-trapping reagent was 1,3,5-tri-*tert*-butyl-2-nitrosobenzene (TTBNB), which was purchased from Tokyo Chemical Industry Co., Ltd. (Tokyo, Japan); HFiP and methanol were purchased from Fluorochem Ltd. (Derbyshire, UK) and FUJIFILM Wako Pure Chemical Co. (Osaka, Japan), respectively.

### 2.2. Sample Preparation

PA66 purified by reprecipitation and the spin-trapping agent were dissolved in HFiP and cast on a glass plate. After air-drying, the residual solvent in PA66 was removed in a vacuum dryer at 50 °C for 3 h. The concentration of TTBNB in the PA66 matrix was controlled at 3 wt%. The thickness of the film was ca. 500 μm.

### 2.3. Characterization

#### 2.3.1. TG-DTA Measurements

Weight loss and differential thermal analysis versus temperature were obtained using a thermal gravimetric-differential thermal analysis (TG-DTA) apparatus (TG/DTA6200, Hitachi High-Tech Science Co. (Tokyo, Japan)). Approximately 5 mg of the PA66 sample was placed on an aluminum pan and heated from room temperature (RT) to 600 °C at a rate of 10 °C/min under a nitrogen gas flow of 50 mL/min.

#### 2.3.2. GPC Measurements

Molecular weight (MW) distribution changes in the PA66 were measured before and after heating, using a gel permeation chromatography (GPC) system (HLC-8320GPC, Tosoh Co. (Tokyo, Japan)) with three columns of TSKgel (GMH_HR_-H(S), Tosoh Co. (Tokyo, Japan)) connected in series. MWs were calibrated using poly(methyl methacrylate) standards. The PA66 samples were dissolved in HFiP solution and then filtered using a 0.45 μm-sized pore membrane. The injected sample concentration of PA66 was 0.1 wt%. In each measurement, a sample of 10 μL was injected at a flow rate of 0.175 mL/min. The temperatures of the columns and the RI detector were maintained at 40 °C continuously. The number average MW, *M*_n_, and the weight average MW, *M*_w_, were calculated from the distribution curve.

#### 2.3.3. FTIR Measurements

Fourier-transform infrared spectroscopy (FTIR) was performed using a transmission method (IR-670, Agilent Technologies, Inc. (Santa clara, CA, USA)). All the spectra were recorded with the wavenumber in the range 4000 cm^−1^ to 400 cm^−1^ at a resolution of 2 cm^−1^ by averaging 256 scans. A thin PA66 film of approximately 10 μm for FTIR was prepared by casting method using HFiP.

#### 2.3.4. ESR Measurements

ESR measurements were carried out on an ESR spectrometer (E500, Bruker Co. (Billerica, MA, USA)) using an X-band microwave at a frequency of approximately 9.5 GHz and a power of 1.0 mW. The magnetic field was around 3300 G, and the modulation width was 1 G at 100 kHz. Cr^2+^ was used as an external standard for radical amounts. The 70 mg of PA66 with TTBNB was placed and sealed in a 5 mm diameter quartz ESR test tube under vacuumed condition. The heating temperature was controlled precisely from RT to 240 °C with a thermal controller unit (DVT3000, Bruker Co. (Billerica, MA, USA)). The obtained ESR spectra were carefully simulated, separated, and qualified using a personal computer and commercial software (Igor Pro, WaveMetrics Inc. (Portland, OR, USA)). The multiple peak ESR spectra were simulated using mixed Gaussian–Lorentzian functions for all peaks with different spectral parameters, including the center field, the integrated intensity of the peak area, the peak-to-peak width, and the Gaussian–Lorentzian ratio.

## 3. Results and Discussion

### 3.1. TG-DTA of PA66

The thermal stability of PA66 was evaluated by measuring weight loss, as well as heat flow in and out using TG-DTA at a heating rate of 10 °C/min under an inert nitrogen atmosphere. As shown in Figure 2, there was a weight loss of about 8% at 250 °C in the TGA curve. This loss is not attributed to thermal degradation but mainly to the volatilization of water and low molecular weight components generated during polymerization because the DTA curve shows only a featureless, endothermic curve over this temperature range. The endothermic peak at 261 °C in the DTA curve is attributed to the melting of crystalline PA66 [29]. Although significant thermal decomposition can be observed from 350 °C to 450 °C in the TGA curve, no peak and no apparent curvature change due to the thermal decomposition is observed below 330 °C, which includes the temperature range in which ESR measurement was performed. DTA showed an endotherm peak between 350 °C and 450 °C; this is considered to be due to a decrease in the heat capacity of the sample and the evaporation of volatile components [21] caused by rapid weight loss from thermal decomposition.

### 3.2. Thermal Degradation Analysis with GPC

The MW of PA66 was compared before and after heating using the same stepwise heating profile of ESR measurement under vacuum conditions. The *M*_n_, *M*_w_, and the degree of dispersion (*M*_w_/*M*_n_) are summarized in Table 1. Degradation did not progress significantly, even when the sample was heated at a temperature below 240 °C; this is thought to be because of the short heating time and the lack of oxygen. However, there was a slight decrease in *M*_n_, and MW dispersion was slightly greater after heat treatment. These changes suggest that the results of the ESR measurements in the latter section show the reactions in the very early stages of thermal degradation.

### 3.3. Thermal Degradation Analysis with FTIR

FTIR analysis was performed to investigate chemical structural changes caused by thermal degradation at a low temperature range below 200 °C. In the ESR analysis described in the next section, each ESR spectrum was taken over 4 min during stepwise heating of the sample, for 50 min until the temperature reached 240 °C. However, because a change of at least a few percentage points was required to detect a distinct difference between FTIR spectra before and after degradation, the PA66 samples had to be heated for a longer time, of the order of 10 h. Figure 3 shows the FTIR spectra of PA66 before and after heating at 160 °C for 13 and 82 h under vacuum conditions. It can be seen that there is little noticeable degradation in progress. However, a detailed comparison reveals a slight increase in amino groups in the 3300 cm^−1^ region. On the other hand, there are no significant changes in the region of carbonyl groups in the 1500–1700 cm^−1^ region, suggesting that, while peroxidative degradation did not proceed because of the non-oxidative condition, main chain scission between the amide group and the adjacent carbons, −CO−NH−CH_2_−, may have occurred, resulting in the formation of secondary amines.

### 3.4. Radical Analysis by Spin-Trapping ESR

#### 3.4.1. ESR Measurements

The results of ESR measurements of PA66, including the spin-trapping reagent TTBNB sample with stepwise heating at 20 °C intervals, are shown in Figure 4. A slight ESR signal can be observed at the center from 80 °C, and it increases with increasing temperature with other signals; the spectral intensity finally reaches a maximum at 220 °C. Then, the signals suddenly almost disappear at 240 °C. Similar measurements on PA66 without TTBNB did not show any ESR signals. Therefore, the signals observed in Figure 4 were spin adducts resulting from the trapping of radical intermediates generated from PA66 as a result of thermal degradation. To separate the spin adducts and to determine from which radicals they originate, computational simulation analysis for all spectra observed in Figure 4 was performed.

#### 3.4.2. Assignments of Spin Adducts

After elaborate trial and error analyses of the estimation of the produced radical species and the assumption of their spectral parameters, it was concluded that the spectrum consists of a maximum of six spin adduct components. Figure 5 shows an example of a simulation analysis of a spectrum heated to 160 °C, in which all six components can be observed. The assignments of the spin adduct radicals and their hyper coupling constant (*hfcc*) and *g*-values are shown in Table 2, together with the literature references for assignment.

**S_1_**, which shows a 9-line feature, indeed consists of 27 lines with overlapping constructed by the interaction of three types of atoms: one nitrogen, two hydrogen on carbon, and two aromatic *m*-hydrogens. **S_1_** was assigned a nitroxide-type spin adduct derived from ^●^CH_2_−CH_2_−, which was produced by main chain scission [30]. It should be noted that this spin adduct should have at least two consecutive CH_2_ groups because, if the substituent adjacent to ^●^CH_2_− is an amide group NH or a carbonyl group C=O, the value of *hfcc* is a little smaller compared with a simple hydrocarbon group [31].

**S_2_** consists of 18 lines derived from one nitrogen, one hydrogen on carbon, and two aromatic *m*-hydrogen atoms. **S_2_** and was assigned to a nitroxide-type spin adduct generated by trapping −^●^CH− produced by hydrogen abstraction from the methylene chain [32]. Given the value of *a*_H_, the groups on both sides of −^●^CH− were considered to be methylene groups.

**S_3_** has a 12-line structure, consisting of 18 lines generated by trapping −^●^CH− as well as **S_2_**; however, **S_3_** was assigned to an anilino-type. The *hfcc*s of both β-hydrogen and *m*-hydrogens were almost equivalent. The *hfcc* of this anilino-type adduct is known to be unaffected by the chemical structure of the adjacent group of −CH−. The neighboring group has three candidates: methylene, or carbonyl, or amine in the amide group, but, as discussed below, it is most likely to be an amine [31].

**S_4_** with a nine-line structure was assigned to an anilino-type spin adduct produced by trapping the tertiary butyl carbon radical ^●^C(CH_3_)_3_ derived from the thermal degradation of TTBNB [33]. This spin adduct always appears when TTBNB is used and heated over 150 °C and exhibits a low *g*-value of 2.0033, which is easily distinguishable from the other components. **S_5_** was assigned to a self-trapping spin adduct derived from ^●^CH_2_− of the methyl group of TTBNB [16]. The component appears to show a 5-line structure with overlapping, but actually consists of 27 lines and is observed only at a high temperature above 160 °C.

**Table 2 polymers-14-04748-t002:** The molecular structure of the assigned spin adducts, **S_1_**–**S_6_**, and their ESR spectral parameters observed for PA66 with TTBNB heated from RT to 240 °C, as well as the literature referred to in assigning the spin adducts.

*Observed Spin Adducts in This Paper*				*Referred Assignments in Previous Articles*				
	Assigned Molecular Structure	*hfcc* and *g*		Trapped Radical	*hfcc* and *g*		Materials	Ref.
S_1_	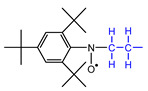	*a*_H_(2)	1.75	^●^CH_2_CH_2_−	*a*_H_(2)	1.799	(*n*-Bu)_3_Sn-Br(or I) /TTBNB/benzene	[30]
		*a* _N_	1.40		*a* _N_	1.346		
		*a* _Hm_	0.08		*a* _Hm_	0.083		
		*g*	2.0053		*g*	2.0060		
S_2_	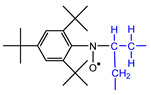	*a*_H_(1)	2.17	−^●^CH−CH_2_−	*a*_H_(1)	2.16	*n*-C_12_C_26_(paraffin) /TTBNB	[32]
		*a* _N_	1.43		*a* _N_	1.33		
		*a* _Hm_	0.08		*a* _Hm_	N/A		
		*g*	2.0046		*g*	2.0061		
S_3_	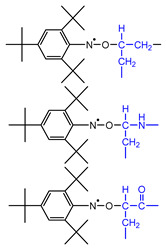	*a*_H_(1)	0.20	−CH_2_−^●^CH−CH_2_−	*a*_H_(1)	0.18	C_6_H_12_/(*tert*-BuO)_2_/TTBNB/benzene	[30]
		*a* _N_	1.00		*a* _N_	1.10		
		*a* _Hm_	0.20		*a* _Hm_	0.18		
		*g*	2.0039		*g*	2.0036 ~ 2.0040		
S_4_	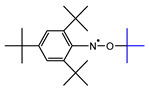	*a* _N_	1.02	^●^C(CH_3_)_3_(TTBNB)	*a* _N_	1.05	*n*-eicosane /TTBNB	[33]
		*a* _Hm_	0.17		*a* _Hm_	0.193		
		*g*	2.0033		*g*	2.0046		
S_5_	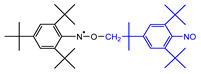	*a*_H_(2)	1.42	^●^CH_2_− (TTBNB)	*a*_H_(2)	1.46	PP/TTBNB	[16]
		*a* _N_	1.29		*a* _N_	1.05		
		*a* _Hm_	0.08		*a* _Hm_	0.10		
		*g*	2.0052		*g*	2.0054		
S_6_	*unassinged*	*a* _H_	N.A.		*a* _H_	-	N.A.	
		*a* _N_	1.3~1.5		*a* _N_	-		
		*a* _Hm_	N.A.		*a* _Hm_	-		
		*g*	2.0033		*g*	-		

**S_6_**is a broad component consisting of more than five peaks, with no clear hyperfine structure observed by wide broadening. **S_6_** is probably a macro-radical with low molecular mobility, which is fixed at the interface between the crystalline and amorphous phases of PA66. Since this broad component was also apparent in the spectrum for the sample heated at 80 °C, it may be a degradation sub-product radical originating from peroxides which have been produced before ESR measurement. The *g*-value of **S_6_** is 2.0033, and the *hfcc* of about 1.3–1.5 mT and 5–7 lines can be roughly read out from this structure. This is most likely a spin adduct derived from an alkyl radical. However, the true structure of **S_6_** is currently unknown; it was not possible to observe a more detailed *hfcc* for this component.

### 3.5. Reaction Mechanism of PA66 Thermal Degradation

This discussion of degradation mechanisms is based on ST-ESR and other analytical results. First, it should be noted that the temperature range, from RT to 240 °C, used in this study was relatively low compared with previous studies of PA66. There is no significant weight loss up to 240 °C, as shown by TG-DTA analysis (Figure 2), indicating that there is no production of volatile compounds, such as cyclopentanone or CO_2_, in this temperature range as would occur with higher temperature pyrolysis [20]. This finding is confirmed by FTIR measurements (Figure 5) where no apparent spectral change is observed, other than a slight increase in amino group. From recent studies [25,34], it has been proposed and accepted that the initiation reaction in thermal oxidative degradation at 90–200 °C is a hydrogen abstraction (detachment) from the α-carbon methylene group next to the NH group, as shown in Figure 1. This mechanism is based on the fact that the bond dissociation energy of C−H in the methylene group next to NH (355–360 kJ/mol) is lower than that of C−H in other methylene units (395 kJ/mol). If this methine radical (−NH−^●^CH−) is trapped by TTBNB, the **S_3_** component will be produced. Figure 6 shows the temperature dependence of radical generation for each spin adduct component. **S_3_** is generated from a low temperature of 100 °C and increases from 160 °C up to 200 °C. If oxygen is present around the PA66, the oxygen quickly adds on the carbon radical −^●^CH− and subsequent degradation reactions via peroxidation and homolysis proceed, eventually cleaving the NH−CH_2_ bond to produce secondary amines and aldehydes. However, if no oxygen is present, as in the case of our ESR measurements, −NH−^●^CH− may directly proceed to two types of main chain scission by β-scission reactions: one produces ^●^CH_2_− radicals and the other produces −^●^C=O radicals, which were not detected by ST-ESR in this paper.

For the production of the secondary amine radical in the thermal degradation of polyamide, two mechanisms have been proposed for temperatures higher than 300 °C, which is much higher than the temperature conditions in our case: either main chain scission producing ^●^NH− and ^●^CH_2_−, or hydrogen abstraction from the β-methylene group to the NH group producing ^●^NH− and −^●^CH− [17], as shown in Figure 2. It has been reported that the bond dissociation energy on the bond NH−CH_2_ is lowest in the PA66 molecular structure [35]; this leads to the assumption that main chain scission at −NH−CH_2_− is more likely in decomposition. Indeed, we observed a large amount of **S_1_** derived from ^●^CH_2_−, as shown in Figure 6. In the case of the β-scission scheme, the main chain radical −^●^CH− is produced, which may be trapped and observed as the **S_2_** spin adduct. However, in our study, we could not detect a spin adduct derived from the amino radical ^●^NH−; this may be due to the high reactivity of the amino radical ^●^NH− and low reactivity of TTBNB against a nitrogen radical.

## 4. Conclusions

The radical mechanism of thermal degradation of polyamide 66 (PA66) occurring under vacuum conditions at temperatures between 80 °C to 240 °C (which includes the temperature range of practical applications), was investigated mainly using a ST-ESR technique. Degradation analysis by TG-DTA on temperature ramp mode and FTIR after heating at 160 °C for 82 h shows no significant weight loss and no sign of oxidative degradation in this temperature range. However, a slight production of secondary amine groups was confirmed by FTIR. GPC analysis showed a small increase in molecular weight dispersion with a small decrease in average molecular weight. Changes in the molecular weight distribution indicate that main chain scission is the prevalent mechanism. ST-ESR analysis reveals that two intermediate radicals are produced via the thermal degradation of PA66: (a) a ^●^CH_2_− radical generated by main chain scission and (b) a −^●^CH− radical generated by hydrogen abstraction from the methylene group of the main chain. The ST-ESR did not directly confirm that a −NH−^●^CH− radical is produced by hydrogen abstraction or detachment, although this reaction had been inferred before as the initiation reaction of thermal degradation of PA at the relatively low temperature below 200 °C. However, the ESR analysis reveals the presence of −^●^CH− radicals and strongly supports the occurrence of this reaction which takes place on the α-carbon next to the NH group. As we show in this paper, the ESR analysis reveals a very small level of reactions that cannot be observed by common analytical methods, such as FTIR and NMR. As a next step, the elucidation of radical degradation mechanisms of PA in the presence of oxygen is also needed, and this work is currently underway.

## Data Availability

Not applicable.

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
