# Peer review of "Spin-Trapping Analysis of the Thermal Degradation Reaction of Polyamide 66"

_polymers, 2022, doi:10.3390/polym14214748_

Round 1

Reviewer 1 Report

In the submitted paper Thermal Degradation Reaction of Polyamide 66 were studied by Spin-Trapping Analysis. Overall, the work is good. However, there are some minor issues that need to be addressed. On the page 3, lines 131-134 authors are describing FT-IR measurements. Although they mentioned transmission, they missed to emphasize by which method exactly; perhaps Attenuated total reflection (ATR)? Page 4, lines 153-154 authors stated: “This loss is not attributed to thermal degradation but mainly to the volatilization of water…” Can you explain the origin of this water; structural or during preparation stage? The same page, next lines 154-156, authors are attributing endo peak on the DTA curve to the melting of PA66. This could be easily confirmed with the DSC analysis. Authors are advised to proceed with the DSC analysis. The same page, lines 160-161, authors correlate the “decrease in the heat capacity of the sample and the evaporation of volatile components”. Could they assume which components? Likewise, one the Fig. 2 authors showed TG and DTA curves of PA66. Where is the DTG curve? It would be very useful during commenting on the thermal degradation if you have corresponding DTG curve. By reading this section, in order to clarify the situation under 200 °C, it is obvious that you need to measure degradation via Thermogravimetric analysis coupled to Fourier transform infrared spectroscopy or Mass Spectroscopy. This is an idea for the future work; please consider it. Page 5, lines 184-185, authors compared FT-IR spectra and revealed a slight increase in amino groups in the 3.300 cm-1 region. Firstly, did they mean on the peak intensity? Secondly, can authors give more (numeric) data about this “slight increase”? It is not clearly visible on the Fig. 3. This is in direct correlation with the conclusion remark on the page 10, lines 315-316.

Reviewer 2 Report

The manusript is relatively new and interesting. Some modifications should done to if the paper meets the requirement for publicaiton.  Some figures were missing or lack of explanations. Some Figures can be plotted in more colorful mode. The simulation method should be discussed briefly. The relavent references should be updated and the introduction section shoud be improved to further highlight the authors' purpose.  Here are some detailed comments,

1. In Figure 3, the locally enlarged details should be included in Figure (a) in proper locations to reduce the numbers of Figures.

2.t is necessary to give a brief description of simulation. There was a obvious difference of the curve patterns between Figure 5(a) and (b), more detailed analysis is necessary.  In fact,  the difference between Figure 5 (a) and (b) should be done by subtraction to see the precision of the simulation method. 

3. Fiure 6 was missing. No explanations were found for Figure 7. How to construct Figure 7  should be added. Different labels should be used to identify different components  in Figure 7 in case of white black version.

Reviewer 3 Report

This paper reports study of degradation mechanism of PA66 using ST-ESR technique. Results are good but draft of manuscript was prepared carelessly. It must be given care.

It is hard for me to recommend this paper for publication.

Some more points are given below.

In line 13 please replace “vacuum at temperatures between 80 °C to 240 °C” with “

 Vacuum at temperature range between 80 °C to 240 °C”

In line 15 and 16, it is claimed that no significant oxidative degradation was observed. But in line 86, it is reported that this degradation occurs. It will be better if author can look into this.

I think there is no need of Figure-1 as these structures are very well know. Figure-1 may be removed from the manuscript.

FT-IR should be replaced with FTIR.

Figures are in very poor and annoying form. All the Figures must be reformed, draw in a good and an attractive way.

In TGA, initial weight loss of 8 % at 250 °C was attributed to removal of water. In such TGA analysis, removal of water and other residual organic solvents occurs below or at 200 °C. The claim of water removal at 250 °C must be proved with strong literature support.

In line 168 and 169, it is mentioned that Mn, Mw, and PDI are given in Table-1. However, these values are missing in Table-1.

In line 178 and 179, exact time must be mentioned.

Absorption bands must be labeled in FTRI spectra. Also, Figure 3a, b, and c must be of equal size.

Round 2

Reviewer 3 Report

Accepted